# Molecular detection of multidrug resistance pattern and associated gene mutations in *M. tuberculosis* isolates from newly diagnosed pulmonary tuberculosis patients in Addis Ababa, Ethiopia

**Melaku Tilahun** [1,2]☯*, **Ezra Shimelis** [1,3]☯*, **Teklu Wogayehu** [2]‡, **Gebeyehu Assefa** [1], **Getachew Wondimagegn** [4], **Alemayehu Mekonnen** [5], **Tsegaye Hailu** [1], **Kidist Bobosha** [1], **Abraham Aseffa** [1]‡

1 Armauer Hansen Research Institute (AHRI), Addis Ababa, Ethiopia, 2 Department of Biology, Arba Minch University (AMU), College of Natural Sciences, Arba Minch, Ethiopia, 3 School of Public Health, College of Health Sciences, Addis Ababa University (AAU), Black Lion Hospital, Addis Ababa, Ethiopia, 4 KNCV Tuberculosis Foundation, Addis Ababa, Ethiopia, 5 Ethiopian Public Health Association, Addis Ababa, Ethiopia

☯ These authors contributed equally to this work.
‡ These authors also contributed equally to this work.
* mtilahun600@gmail.com (MT); ezrashimelis@yahoo.com (ES)

## Abstract

### Introduction

Multi-drug resistance is a major challenge in the control of tuberculosis. Despite newer modalities for diagnosis and treatment, people are still suffering from this disease. Under-standing the common gene mutations conferring rifampicin and isoniazid resistance is crucial for the implementation of effective molecular tools at local and national levels. Hence, this study aimed to evaluate the molecular detection of rifampicin and isoniazid-resistant gene mutations in *M.tuberculosis* isolates in Addis Ababa, Ethiopia.

### Method

Health Center-based cross-sectional study was conducted between January and September 2017 in Addis Ababa, Ethiopia. The collected sputum samples were processed for mycobacterial isolation and Region of difference 9 based polymerase chain reaction for species identification. To characterize the rifampicin and isoniazid-resistant *M. tuberculosis* isolates, a molecular genetic assay (GenoType MTBDR*plus*) was used; the assay is based on DNA-STRIP technology.

### Result

Culture positivity was confirmed in 82.6% (190/230) of smear-positive newly diagnosed pulmonary tuberculosis cases enrolled in the study. From 190 isolates 93.2% were sensitive for both rifampicin and isoniazid, and 6.8% of the isolates were resistant to at least one of the

**Data Availability Statement:** All relevant data are within the paper and its Supporting Information files.

**Funding:** This work is supported by AHRI core budget. AHRI Receives core support from Sida, Norad and the Government of Ethiopia.

**Competing interests:** The authors have declared that no competing interests exist.

tested anti-TB drugs. Gene mutations were observed in all studied multidrug resistance-associated gene loci (*rpoB*, *katG*, and *inhA*). Two isolates exhibited heteroresistance, a mutated, as well as wild type sequences, were detected in the respective strains. MDR-TB case was observed in 1.1% (2/190) of the cases. All the MDR-TB cases were positive for HIV and found to have a history of prior hospital admission.

## Conclusion

In our finding a relatively high prevalence of any drug resistance was observed and the overall prevalence of multidrug-resistant tuberculosis was 1.1%.The majority of drug-resistant isolates demonstrated common mutations. Heteroresistant strains were detected, signaling the existence of an *M.tuberculosis* population with variable responses to anti-tuberculosis drugs or of mixed infections.

## Introduction

Tuberculosis (TB) is one of the most ancient diseases of mankind, caused by *M.tuberculosis* complex (MTBC). Despite newer modalities for diagnosis and treatment of TB, people are still suffering from this disease. In addition to drug sensitive-TB, multidrug resistance is a major challenge in the control of TB [1]. According to the World Health Organization (WHO) anti-TB drug resistance surveillance data, 3.5% of new and 18% of previously treated TB cases in the world are estimated to have multidrug-resistant (MDR) or rifampicin-resistant (RR) TB [2]. African countries south of the Sahara including Ethiopia are heavily affected by TB and drug resistance TB (DR-TB). Ethiopia notified 125,836 new TB cases in 2016. Amid the notified, 2.7% of new and 14% previously treated TB cases were estimated to harbor DR-TB. In Ethiopia, culture and drug susceptibility testing (DST) is not done regularly for all incident TB cases. The country is underperforming in ensuring early diagnosis and a proportion of cases remain undiagnosed as well continue to transmit the disease in the community [3].

Based on evidence and expert opinion, the WHO endorsed the use of molecular tools for the rapid screening of patients at risk of MDR-TB. Rapid tests can provide results within days and thus allow early and appropriate treatment to decrease morbidity and mortality as well as to interrupt transmission. Among these, line probe assay (LPA) has been developed for the rapid and simultaneous detection of the MTBC and its resistance to RIF and INH [4]. This assay detects for the absence and/or presence of wild type (WT) and/or mutant (MUT) DNA sequences within a specific region of three genes: the *rpoB* gene (RIF resistance) the *katG* gene (high-level INH resistance) and the promoter region of the *inhA* gene (low-level INH resistance) [5].

An understanding of the MDR/RR prevalence of *M. tuberculosis* is critical for effective control of the global burden of TB which is caused by the organism belonging to the MTBC [6]. Likewise studying the genetic mutations that confer RIF and INH resistance in *M. tuberculosis* strains has indispensable importance in the control strategy. Studies were conducted in different parts of Ethiopia to determine the significance and burden of the existing problem. Mekonnen *et al* described multidrug-resistant and heteroresistant *M.tuberculosis* and associated gene mutations in the country [7]. A similar study conducted in Southwest Ethiopia identified the drug resistance-conferring gene mutations in *M.tuberculosis* isolated from pulmonary tuberculosis (PTB) patients [8]. Likewise, a study conducted by Alelign *et al*

reported the existence of a high proportion of drug-resistant *M.tuberculosis* strains from South Gondar Zone, northwest Ethiopia [9]. All these studies emphasized the significance of the drug resistance-conferring gene mutations in *M.tuberculosis* isolates from TB patients.

A study conducted in Saint Peter's TB Referral Hospital between 2015 and 2016 showed that 5.2% of MDR-TB cases were identified among newly diagnosed PTB cases. All the DR-TB cases exhibited drug resistance-conferring gene mutations [10], which indicate the magnitude of the problem in Addis Ababa, Ethiopia. Drug resistance mutations are diverse and may change over time as different clones expand or shrink, and these changes need to be monitored over time. Therefore, this particular study was designed to determine the multidrug resistance pattern and to analyze the frequency of gene mutations associated with RIF and/or INH resistance of *M.tuberculosis* strains isolated from newly diagnosed PTB patients in Addis Ababa, Ethiopia.

## Materials and methods

### Description of the study area

An institution-based cross-sectional survey was conducted in Addis Ababa, the capital of Ethiopia. According to the projection of the Central Statistics Agency (CSA) of Ethiopia, in the year 2014, its population was about 3.2 million [11] and a surface area of 540 Sq.km, resulting in a density of 6218 people per Sq.km. The city has 10 sub-cities and 116 *Woredas* (lowest administrative units in Ethiopia) [12]. Concerning health service delivery, there are 47 hospitals, 100 government health centers, 204 higher private clinics, 226 medium private clinics, and 143 lower private clinics [13]. Of the different health facilities under the city administration, purposely 20 health centers were selected (2 from each of the 10 sub-cities). The health centers were selected based on the TB case report of the city administration health bureau. The total sample size was proportionally allocated based on its case report.

### Study population

The study population was all newly registered bacteriologically confirmed PTB patients with age greater than 15 years who were treated in the selected twenty health centers in Addis Ababa, in 2017. The inclusion criteria were age greater than or equal to 15 years with presumptive PTB disease and showed their willingness to participate in the study. All the study participants signed informed consent. The exclusion criteria were patients with severe illness and unable to provide sputum specimens.

### Sputum collection and AFB staining

The data collection and some of the laboratory procedures are schematically *shown in Fig 1*. Sputum samples were collected consecutively until the required number of samples is achieved from smear-positive presumptive PTB patients enrolled in the study. The patients were asked to produce a productive two sputum specimens in wide-mouthed screw-capped containers outside the room in the sputum collection stand on the spot for AFB staining [3].

The collected sputum samples were smeared and Ziehl-Neelsen stained based on the National quality assurance of smear microscopy for TB diagnosis guideline [14]. The stained smears were examined carefully under the oil immersion objective by the facility laboratory technologist and the reading was systematic and standardized to scan at least 100 high power fields before producing a negative result. The collected and smear-positive sputum samples were pooled in a 50ml falcon tube then transported within a day to AHRI TB laboratory by

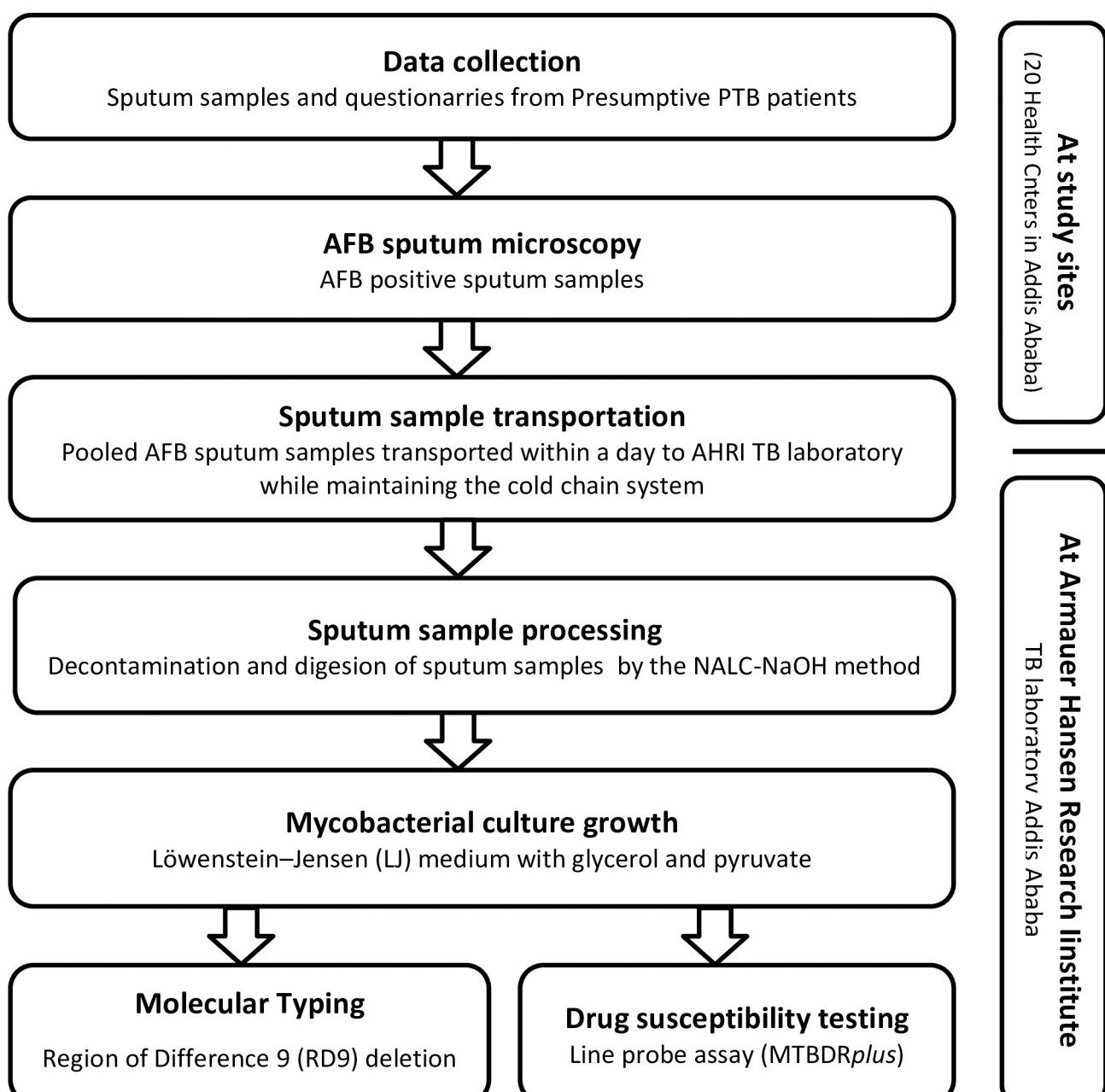

**Fig 1. Flow chart showing the data collection and some of the laboratory procedures.**

maintaining the cold chain system during transportation and further processing took place at AHRI TB Laboratory without delay.

## Mycobacterial culture

Egg-based LJ-pyruvate and LJ-glycerol media were prepared aliquoted and stored in the refrigerator at 2–8°C according to the procedure [15]. Briefly, the sputum samples were decontaminated by the NALC-NaOH method and centrifuged at 3000 rpm for 15 min. The supernatant was discarded and the deposit was resuspended by 1.5 ml phosphate saline buffer solution.

The sediment was inoculated into conventional Löwenstein-Jensen egg medium containing 0.6% sodium pyruvate and glycerol and incubated for at least 8 weeks, with weekly observation for the presence of mycobacterial colonies. Colonies were examined for acid-fast bacilli with Ziehl-Neelsen staining to select AFB-positive isolates. Colonies from AFB positive isolates were collected into two cryovials. One cryovial was used to extract the DNA by heat killing at 80˚C for 1hour according to the procedure [16], for molecular typing and drug sensitivity testing. The other vial was frozen at -80˚C for long term storage of *M.tuberculosis* isolates [15].

## Molecular typing

The heat-killed and DNA extracted isolates were investigated using PCR based deletion typing for the presence or absence of RD9 to differentiate MTBC from other mycobacterial species. The sequences of the primers used for RD9 deletion typing was RD9flankF, 5′-GTG TAG GTC AGC CCC ATC C-3′; RD9intR, 5′-CTG GAC CTC GAT GAC CAC TC-3′; and RD9flankR, 5′-GCC CAA CAG CTC GAC ATC-3′. PCR amplification of the mixtures was performed using a Thermal Cycler PCR machine according to standard procedures [17]. Briefly, the reaction mixture was prepared and amplified using the following program: 10 min at 95˚C for enzyme activation, 1 min at 95˚C for denaturation, 0.5 min at 61˚C for annealing, 2 min at 72˚C for the extension, involving a total of 35 cycles, and a final extension at 72˚C for 10 min. The product was electrophoresed using the Agarose Gel Electrophoresis System in 1.5% agarose gel in 1× Trisacetate-ethylene diamine tetraacetic acid running buffer. Ethidium bromide at a ratio of 1:10, 100 base pair (bp) DNA ladder, and orange 6× loading dye was used in gel electrophoresis and the gel was visualized.

## Drug sensitivity test

To detect the drug sensitivity patterns of 190 culture-positive *M.tuberculosis* isolates, a molecular genetic assay (GenoType MTBDR*plus*) was used. The assay is based on DNA-STRIP technology. Briefly, the whole procedure involves a multiplex PCR amplification with biotinylated primers and reverses hybridization. DNA was extracted from the appropriate culture-positive *M.tuberculosis* isolates. Master Mix was prepared in a cleanroom to prevent contamination. In the sample addition room, the DNA extracts, the positive and negative controls were added to the corresponding PCR tubes. After the addition of the mixture, the tube placed into a PCR machine for amplification. Finally, the amplicon was detected with a series of procedures by adding different reagents to the strip. The strips were formed color bands after the addition of the final substrate reagent. This assay detects for the absence and/or presence of WT and/or MUT DNA sequences within a specific region of three genes: the *rpoB* gene (coding for the β-subunit of the RNA polymerase), for the identification of RIF resistance; the *katG* gene (coding for the catalase-peroxidase), and the promoter region of the *inhA* gene (coding for the NADH enoyl ACP reductase), for the identification of INH resistance. The procedure of the test was performed based on the manufacturer instructions (Hain Life sciences, Nehren, Germany) [18].

## Data analysis

The questionnaires were tested before the actual data collection procedure. Socio-demographic and clinical data obtained through questionnaires and the results of laboratory tests were entered into SPSS data record files. Statistical analysis was performed using version 25 SPSS software package. The Chi-square ($X^2$) test was used to detect statistically significant differences. The adjusted analysis of proportions was made by logistic regression. The odds ratio

was used to measure the degree of association. A probability of < 0.05 was considered significant.

## Ethical considerations

Ethical approval was obtained from the AHRI/ALERT Ethics Review Committee. A support letter was obtained from the Addis Ababa City Administration Health Bureau. The purpose and benefit of the study were explained to each study subject to come up into consensus and willing to participate in the study was signed the informed consent form prepared based on the local language and included in the study. For participants < 18 years of age, assent was obtained, as well as consent from their parent or legal representative. All the drug susceptibility test results were reported to the respective health facilities for further management of the patients.

## Results

### Socio-demographic characteristics

In this study, a total of 260 study participants were enrolled and provided sputum for laboratory analysis. Due to contamination and mislabeling, 30 of them were excluded from the study, and further analysis is based on 230 cases. The overall culture positivity was 82.6% (190/230). The median age of the study participants was 27, while the mean age was 30.43(+/- 11.14 SD) years. Majority 171 (90%) were between the age group of 15 and 44 years. The socio-demographic characteristics of the study participants are shown in Table 1. Of the 190 culture-positive study participants 107 (56.3%) were male and 83 (43.7%) were female, resulting in male to female ratio of 1.3:1.

### *M. tuberculosis* species identification

All the 190 isolates were subjected to RD9 deletion typing with PCR to differentiate *M. tuberculosis* species from other members of the MTBC. *M. tuberculosis* H37Rv, *M. bovis* bacilli Calmette-Guérin (BCG) were included as a positive control and molecular grade water was used as a negative control. Interpretation of the result was based on bands of different sizes (396bp for *M. tuberculosis* and 575 bp for *M. bovis*) as previously described. The test result shown that all isolates had intact RD9 locus and were subsequently classified as *M. tuberculosis* species. No other species of MTBC was detected.

### Drug resistance patterns of *M. tuberculosis* isolates

The drug sensitivity test was performed for all the 190 culture-positive *M. tuberculosis* isolates obtained from newly diagnosed PTB patients. From the tested isolates 93.2%, were fully susceptible to both INH and RIF, and 6.8%, were resistant to at least one of the tested anti-TB drugs as shown in Table 2. The two age brackets 20–24 and 25–29 cumulatively accounted for a higher percentage of (61.5%) the drug-resistant isolates compared with other age brackets. Despite the observed differences in the proportions of drug-resistant cases, no statistically significant association was observed between demographic characteristics of patients with drug resistance. MDR-TB (resistance to both RIF and INH) was observed in 1.1% (2/190) of the cases.

### Clinical presentation and MDR-TB associated risk factors

During the study period majority of culture, positive study participants presented to the health center with a chief complaint of cough 185/190 (97.4%), night sweet 90/190 (47.4%), weight

**Table 1. Socio-demographic characteristics of the study participants, Addis Ababa, 2019 (n = 190).**

| Variables | Frequency in No (%) |
|---|---|
| **Age Groups** | |
| 15–19 | 18(9.5) |
| 20–24 | 47(24.7) |
| 25–29 | 49(25.8) |
| 30–34 | 23(12.1) |
| 35–39 | 16(8.4) |
| 40–44 | 18(9.5) |
| 45–49 | 4(2.1) |
| 50> | 15(7.9) |
| **Gender** | |
| Male | 107(56.3) |
| Female | 83(43.7) |
| **Marital Status** | |
| Married | 73 (38.4) |
| Single | 101(53.2) |
| Widowed | 6(3.2) |
| Divorced | 10(5.3) |
| **Educational Level** | |
| Illiterate | 30(15.8) |
| Read and Write | 17(8.9) |
| Elementary (1–8) | 60(31.6) |
| Secondary School (9–12) | 60(31.6) |
| College and Above | 23(12.1) |
| **Occupational Status** | |
| Farmer | 5(2.6) |
| Civil servant | 24(12.6) |
| Merchant | 51(26.8) |
| Student | 13(6.8) |
| Unemployed | 20(10.5) |
| Housewife | 21(11.1) |
| Daily laborer | 56(29.5) |

loss 89/190 (46.8%), and bloody sputum (hemoptysis) 37/190 (19.5%). MDR-TB associated risk factors of the study participants are summarized in Table 3. Based on the study participants' response, 32/190 (16.8%) had TB patient contact in their household, 26/190 (13.7%) had a history of prior hospital admission and 31/190 (16.3%) had a history of chronic illness (diabetes mellitus, asthma and/or chronic obstructive pulmonary diseases). Any type of habitual

**Table 2. The drug resistance patterns of *M. tuberculosis* isolates, Addis Ababa, 2019 (n = 190).**

| Drug Resistance Pattern | Frequency in No (%) |
|---|---|
| Fully susceptible | 177(93.2) |
| Any resistance | 13(6.8) |
| Monoresistance | 11(5.8) |
| Hetroresistance | 2 (1.1) |
| MDR(INH + RIF resistance) | 2(1.1) |

**Table 3. MDR-TB associated risk factors, Addis Ababa, 2019 (n = 190).**

| Risk Factors | Frequency N (%) | MDR-TB (INH+RIF) | |
|---|---|---|---|
| | | Positive N (%) | Negative N (%) |
| **HIV Result Status** | | | |
| Positive | 26 (16.4) | 2(7.7) | 24 (92.3) |
| Negative | 133 (83.7) | 0 (0) | 133 (100) |
| Unknown | 31 (16.3) | 0 (0) | 31 (100) |
| **History of Any Hospital Admission (Yes/No)** | | | |
| Yes | 26 (13.7) | 2(7.7) | 24 (92.3) |
| No | 164 (86.3) | 0 (0) | 164 (100) |
| **History of TB Patient Contact (Yes/No)** | | | |
| Yes | 32 (16.8) | 0 (0) | 32 (100) |
| No | 158 (83.2) | 2(1.3%) | 156 (98.7) |
| **History of Cigarette Smoking (Yes/No)** | | | |
| Yes | 28 (14.7) | 1(3.6) | 27 (96.4) |
| No | 162 (85.3) | 1(0.6) | 161 (99.4) |
| **Habitual Alcohol Drinking (Yes/No)** | | | |
| Yes | 66 (34.7) | 1(1.5) | 65 (98.5) |
| No | 124 (65.3) | 1(0.8) | 123 (99.2) |
| **History of Chronic Illness (Yes/No)** | | | |
| Yes | 31 (16.3) | 1(3.2) | 30 (96.8) |
| No | 159 (83.7) | 1(0.6) | 158 (99.4) |

From the total culture-positive study participants, 159/190 (83.7%) of them were interviewed and their HIV test result was obtained from the registration book, whereas for 16.3% no HIV test results could be obtained. Among the MDR-TB suspected cases, 26/159 (16.4%) were TB/ HIV co-infected.

alcohol drinking was reported by 66/190 (34.7%) of the study participant while cigarette smoking was reported by 28/190 (14.7%) of the study populations.

## Mutation patterns of drug-resistant *M.tuberculosis* isolates

The frequency of gene mutations associated with resistance to RIF (*rpoB*) and INH (*katG* and *inhA*) is shown in Table 4. The gene mutations were observed in all of the drug resistant-associated gene loci conferring that there was RIF and INH resistance. One of the tested isolates has shown both of the two (*katG* and *inhA*) canonical mutations conferring INH resistance. The absence of the *rpoB* WT8 gene with the corresponding hybridization of *rpoB* MUT3 (S531L substitution) was observed in one of RIF resistant *M. tuberculosis* isolates. While failing of WT7 with the entrance of MUT2A (H526Y substitutions) shared in one of RIF-resistant

**Table 4. Mutation patterns of drug-resistant *M. tuberculosis* isolates, Addis Ababa, 2019, (n = 13).**

| Gene | Mutation pattern(Wild Type/Mutant) | Amino acid change | MDR-TB (n = 2)N (%) | RIF/INH Monoresistance (n = 13)N (%) | Resistance pattern |
|---|---|---|---|---|---|
| *rpoB* | WT7/MUT2A | H526Y | 1 (50) | 0 | MDR |
| | WT8/MUT3 | S531L | 1(50) | 0 | MDR |
| *katG* | WT/MUT1 | $S315T_1$ | 2(100) | 10 (76.9) | Monoresistance |
| | WT+MUT1 | $S315T_1$ | 0(0) | 1(7.7) | Heteroresistance |
| *inhA* | WT1/MUT1 | C15T | 0(0) | 4 (30.8) | Monoresistance |
| | WT1+MUT1 | C15T | 0(0) | 1(7.7) | Heteroresistance |

INH = Isoniazid, MDR = Multidrug resistance, MUT = Mutant Type, RIF = Rifampicin, WT = Wild Type.

gene mutations. Resistance to INH is associated with a mutation at two genes; *katG* and *inhA* promoter region. Among 13 INH-resistant strains, 76.9% (10/13) were due to the absence of the *katG* WT gene with the corresponding hybridization of the *katG* MUT1 probe (Ser315Thr1 substitution) indicating high-level INH resistance. Unexpectedly one (7.7%) isolate exhibited positive hybridization of both WT and corresponding MUT1 (Ser315Thr1 substitution) probe of the *katG* gene signaling hetroresistance pattern and considered as 'rare' mutation. From the INH resistant isolates 30.8% (4/13), were due to failing at the *inhA* WT1 gene (C15T substitution). Similarly, an unforeseen mutation was observed in the *inhA* gene mutation. One isolate (7.7%) revealed positive hybridization of both WT and corresponding MUT1 (C15T substitution) probe of the *inhA* gene signaling hetroresistance pattern and considered as 'rare' mutation. Based on gene mutation analysis, both multidrug-resistant isolates were due to Ser315Thr1 substitution in the *katG* locus. Nonetheless, all of the INH-monoresistant has shown mutation at the *inhA* gene, whereas not any one of the MDR-TB strains revealed *inhA* gene mutation.

## Discussions

In this study, we investigated the multidrug resistance pattern and the associated gene mutation of *M. tuberculosis* isolates from newly diagnosed PTB patients in Addis Ababa, the capital of Ethiopia. The median age of the study participants was 27, while the mean age was 30.43(+/-11.14 SD) years. The majority of the study participants were between the age group of 15 and 44 years, which is the most agile and economically active age group. The fact that this age group is a driving force of the economy of Ethiopia, might suggest that TB is applying a considerable influence on the economy of the country. In supporting our finding a high risk of infection in this age group relates to having a higher number of social contacts in the community during young adulthood [19].

The molecular drug susceptibility test was performed for all the 190 culture-positive *M. tuberculosis* isolates obtained from newly diagnosed PTB patients. From the tested isolates 93.2% were fully susceptible to both INH and RIF, and 6.8% of the isolates were resistant to at least one of the tested anti-TB drugs. This is relatively lower than previous reports from different parts of the country [8, 9]. In a similar manner lower level of INH and RIF resistance was reported from drug naïve TB patients in East Africa [20, 21]. A published review and WHO programmatic management of DR-TB guidelines stated that the variation in the overall prevalence of DR-TB among different study settings could be due to variation in sample size, poor TB case management, underprivileged diagnosis setup and irregular supplies of anti-TB drugs [22, 23]. It can be argued that the time of the study can play a significant role. Drug resistance is a function of clones that circulate everywhere, some more efficiently than others and wax and wane with time.

In our study, INH monoresistance was observed in 5.8% of the isolates. The highest monoresistance to INH drug from newly diagnosed PTB patients were reported from different parts of the country [7–9]. In a retrospective study, a relatively similar INH mono resistance was reported from the National Tuberculosis Reference Laboratory and the seven Regional Laboratories in Ethiopia [24]. The relatively high proportion of INH resistance compared with RIF in our study could be due to the common use of INH for a longer time because it is cheap, effective and has a lower rate of adverse events [25]. Besides, presently INH is used as a preventive therapy for people living with HIV to reduce the risks of active TB development [26]. This condition may result in an increased occurrence of MDR-TB strains if RIF resistance becomes raised. Therefore INH mono resistance should be properly reported and monitored to minimize the spread of MDR-TB strains in the study area.

In our investigation, zero RIF mono resistance was observed from the tested isolates. In agreement with our finding a relatively low level of RIF monoresistance was reported from different parts of the country [9, 27]. High RIF mono resistance (4.9%) was reported from Southwest Ethiopia [8]. The reason for the minimum RIF mono resistance could be the recent addition and control use of RIF in the TB treatment regimen as compared to other first-line anti-TB drugs and the low prevalence of RIF in the country in general, requiring a higher sample size to pick the cases. If the drug monitored properly the recent treatment regimen may be used for a long time with minimal resistance from the bacilli.

The gene mutations were observed in all studied multidrug resistance-associated gene loci (*rpoB*, *katG*, and *inhA*). In the present study we observed mutations at S531L substitution in RIF resistant *M. tuberculosis* isolate, this was the most frequently reported mutation in RIF resistant *M. tuberculosis* isolates in Ethiopia [7, 8, 28]. In our result, H526Y substitution, a rarely reported RIF resistance-conferring gene mutation was identified. In line with our finding infrequent RIF resistance-conferring gene mutations were reported from smear-positive PTB cases in Addis Ababa and somewhere else [10, 29], demanding the need for recurrent monitoring of RIF resistance mutations in the country.

An earlier study in our country indicated that the majority of INH resistance was due to *katG* gene mutations which involved base changes at codon 315 (S315T1 substitution) [7–10]. In agreement with this finding, 76.9% of INH-resistant strains from Addis Ababa have a mutation at the *katG* gene (S315T1 substitution) revealed high-level INH resistance. Mutation in the *inhA* (C15T substitution) promoter region usually reported in INH monoresistant isolates [8, 10]. In line with this result, in our study, 30.8% of mutations in the promoter region of the *inhA* gene (C15T substitution) were found in the INH monoresistant isolates. Therefore, mutations at the *katG* gene were significantly associated with MDR-TB compared to *inhA* gene mutations.

In our study unexpected mutation patterns were detected, both the WT and the corresponding MUT1 probes of the *katG* and *inhA* genes were positive signaling a heteroresistance pattern and considered as 'rare' mutations. These unexpected rare mutations were also observed in similar studies in our country [7, 9]. The heteroresistance pattern might be due to the presence of the *M. tuberculosis* population with variable responses to anti-TB drugs or of mixed infections with sensitive and resistant isolates and it is thought to be a preliminary stage to full resistance [29].

This heteroresistance phenomenon will endanger the effective treatment of patients with INH thereby leading to the development of anti-TB drug resistance in the study area.

The prevalence of MDR-TB in our study was 1.1% somewhat minimal compared with the National MDR-TB prevalence [3]. However, our result was in line with previous studies in our country [8, 9, 30, 31]. High MDR-TB (4.3%) prevalence was reported from the National Tuberculosis Reference Laboratory and the seven Regional Laboratories among newly diagnosed TB patients [24]. This may be due to variation in sample size and studied population. The other possible reason could be recently the government launched an active TB case finding strategy, sample referral system, and the diagnosis capacity of the health facilities are maximized and supported by the GeneXpert machine. For instance, in our study site, the selected 20 Health Centers implemented GeneXpert for TB diagnosis. In these pieces of practical evidence, early detection of RIF resistance strains could be possible. This might result in early referral and treatment of MDR-TB cases contributes to the decreased prevalence in our study settings in particular and in our country in general.

In this study the number of isolates resistant to both RIF and INH (MDR-TB) was two, therefore, we could not demonstrate an overall association between the MDR-TB cases and the studied risk factors, but our result suggests that the MDR-TB cases were positive for HIV

and had a history of prior hospital admission. Our results could be casual however in the previous study significant association was reported [32]. Few works of literature show the direct association of developing MDR-TB with HIV infection and hospital admission. Mesfin *et al.*, in the study indicated that there was a significant association between TB/HIV co-infection, smoking of cigarettes, alcohol drinking, hospital admission, and health facility visiting for developing MDR-TB [33]. A case-control study from St Peter's TB Specialized Hospital showed that there was a significant association between HIV infection and MDR-TB development [34].

The association of anti-TB drug resistance development with HIV infection and prior hospital admission could be explained by the fact that HIV infected patients in developing countries have a rapid disease progression and develop active TB infections much faster than immune-competent people [35]. The situation gets worse in settings where MDR-TB is prevalent, either in the general population or in the local population such as a hospital. Patients infected with MDR-TB require longer, more expensive treatment regimens than drug-susceptible TB, with poorer treatment success [36]. Furthermore, people living with HIV may also be more likely to be exposed to MDR-TB patients, due to either increased hospitalization in settings with a poor infection control or association with peers who may have MDR-TB. For the association, the other possible reason could be poor treatment adherence because HIV/AIDS patients may take a combination of various treatments including antiretroviral therapy this condition may lead to not attending their anti-TB drugs due to the adverse effect or boredom [37]. This may lead to the rapid development of a pool of DR-TB patients. Besides, drug malabsorption in HIV infected patients can also lead to drug resistance and has been shown to result in treatment failure. This condition may lead to the accumulation of drug-resistant strains [38]. In general, the HIV epidemic serves as an amplifier of TB outbreaks by providing a reservoir of susceptible hosts [39], the collective effect may result in the increase and transmission of MDR-TB strains in the general population.

There are two major limitations in this study that could be addressed in future research. First, the study focused only on the molecular detection of INH and RIF resistance-conferring gene mutations in *M.tuberculosis* strains from newly diagnosed PTB patients. We didn't perform the conventional DST for RIF and INH resistant *M. tuberculosis* isolates. The second limitation of our study is the low prevalence of RIF resistance in our findings. This study might not be representative of the whole population and geographical area of the study site. A large scale study with a higher sample size to pick the cases will be required.

## Conclusions and recommendations

In our finding a relatively high prevalence of any one drug resistance and INH mono resistance was observed; the overall prevalence of MDR-TB was 1.1%. The majority of drug-resistant isolates demonstrated common mutations. Heteroresistant strains were detected, signaling the existence of the *M.tuberculosis* population with variable responses to anti-TB drugs or of mixed infections. MDR-TB is more prevalent in HIV positive patients and those who had a history of prior hospital admission. Therefore, in Ethiopia sequence-based genetic analysis of drug resistance *M.tuberculosis* is required to further understand a wide range of gene mutations and to improve the existing molecular tools to capture mutations that do not appear in the current test.

## Supporting information

**S1 File.**
(SAV)

## Acknowledgments

The author would like to acknowledge Addis Ababa City Administration Health Bureau, the 20 Health Center Laboratories, TB Clinic staff, and health professionals for their great help in providing the necessary cooperation during sample collection. They would also like to present special thanks to the study participants (partners) who were collaborative to be part of this project by providing the necessary information and sputum specimens, without their participation this project is worthless.

## Author Contributions

**Conceptualization:** Melaku Tilahun, Ezra Shimelis.

**Data curation:** Melaku Tilahun, Tsegaye Hailu.

**Formal analysis:** Melaku Tilahun, Ezra Shimelis, Tsegaye Hailu, Abraham Aseffa.

**Investigation:** Melaku Tilahun, Ezra Shimelis, Gebeyehu Assefa.

**Methodology:** Melaku Tilahun, Ezra Shimelis, Gebeyehu Assefa, Alemayehu Mekonnen, Abraham Aseffa.

**Project administration:** Melaku Tilahun, Gebeyehu Assefa, Getachew Wondimagegn, Kidist Bobosha.

**Resources:** Melaku Tilahun.

**Software:** Tsegaye Hailu.

**Supervision:** Teklu Wogayehu, Getachew Wondimagegn, Alemayehu Mekonnen, Kidist Bobosha, Abraham Aseffa.

**Validation:** Alemayehu Mekonnen, Tsegaye Hailu.

**Visualization:** Melaku Tilahun.

**Writing – original draft:** Melaku Tilahun.

**Writing – review & editing:** Ezra Shimelis, Teklu Wogayehu, Kidist Bobosha, Abraham Aseffa.

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
