## [Decision Letter · Decision Letter 0]

16 Apr 2020

PONE-D-20-08561

Molecular detection of rifampicin and isoniazid-resistant gene mutation in M.tuberculosis isolates from newly diagnosed pulmonary tuberculosis patients in Addis Ababa, Ethiopia.

PLOS ONE

Dear Dr. Tilahun,

Thank you for submitting your manuscript to PLOS ONE. After careful consideration, we feel that it has merit but does not fully meet PLOS ONE’s publication criteria as it currently stands. Therefore, we invite you to submit a revised version of the manuscript that addresses the points raised during the review process.

We would appreciate receiving your revised manuscript by May 31 2020 11:59PM. To enhance the reproducibility of your results, we recommend that if applicable you deposit your laboratory protocols in protocols.io, where a protocol can be assigned its own identifier (DOI) such that it can be cited independently in the future. For instructions see: http://journals.plos.org/plosone/s/submission-guidelines#loc-laboratory-protocols

We look forward to receiving your revised manuscript.

Kind regards,

HASNAIN SEYED EHTESHAM

Academic Editor

PLOS ONE

Journal Requirements:

2. We note that you included minors (age<18) in your study. Please provide additional details regarding minors consent. In the ethics statement in the Methods and online submission information, please ensure that you have specified whether you obtained consent from parents or guardians. If the need for consent was waived by the ethics committee, please include this information.

3. Your ethics statement must appear in the Methods section of your manuscript. If your ethics statement is written in any section besides the Methods, please move it to the Methods section and delete it from any other section. Please also ensure that your ethics statement is included in your manuscript, as the ethics section of your online submission will not be published alongside your manuscript.

Additional Editor Comments (if provided):

I have gone through the manuscript and also the comments of the two experts. While one recommends major revision, the other recommends rejection for the reason being the results are geography specific. I would like to overrule this Reviewer.

Reviewers' comments:

Reviewer's Responses to Questions

**Comments to the Author**

1. Is the manuscript technically sound, and do the data support the conclusions?

Reviewer #1: Yes

Reviewer #2: Partly

2. Has the statistical analysis been performed appropriately and rigorously? 

Reviewer #1: Yes

Reviewer #2: Yes

3. Have the authors made all data underlying the findings in their manuscript fully available?

Reviewer #1: Yes

Reviewer #2: No

4. Is the manuscript presented in an intelligible fashion and written in standard English?

Reviewer #1: Yes

Reviewer #2: No

5. Review Comments to the Author

Reviewer #1: The study reports on the resisance patterns of newly diagnosed M. tuberculosis and the usefulness of molecular detection of rifampicin and isoniazid resistance gene mutations in Addis Abeba in Ethiopia. Studies from other regions do exist already and are discussed in the discussion section. The present study is very specific to this geographical region and yields results of importance for this region, therefore I believe that the manuscript is perhaps less suitable for the broad readership of Plos One but might be better placed in a journal more specific for the region concerned.

Language corrections are required.

Some minor points:

• Introduction, line 70 (MDR/RR-TB, MDR-TB):

o Please define how exactly MDR/RR-TB and DR-TB are defined and separated from each other.

• Materials and Methods, line 119 (“the guideline”):

o Which guideline do you mean?

• Materials and Methods, line 123 (“pooled samples”):

o What are the pooled samples used for?

• Table 4:

o The ODDS ratio for the HIV Positive is missing.

Reviewer #2: The present study deals with the characterization of drug-resistant patterns of newly identified and diagnosed TB cases in various parts of Addis Ababa, Ethiopia. Ethiopia is among the 30 high burden TB endemic region. Effective monitoring and management of TB cases are urgently required for the containment of the TB disease in high burden regions. The authors studied the resistant pattern in 190 cases of newly diagnosed TB. In the present study, the authors identified two risk factors that are associated with MDR-TB development i.e., chronic illness and HIV infection. Although, earlier reports mentioned higher INH mono-resistance or INH/RIF mono resistance prevalence in the regions of Ethiopia, one study has mentioned a similar pattern of mono resistance. The observed higher resistance cases of INH resistance maybe due to the prevalent use of INH as preventive therapy of TB in HIV infected cases in the region studied. Most of the resistance was due to the common mutations occurring in the INH as well as RIF gene. The authors also observed hetero-resistance cases to the INH, which might be due to the mixed isolates infection. The authors also find a low occurrence of MDR-TB cases in the study population, which has been reported to have higher percentage of MDR-TB cases. Overall, the manuscript cannot be considered for publication in present form and will require substantial improvement for consideration. The primary concern regarding the current manuscript is the sample size. The overall sample size is low and can affect the outcome of the study. The following queries may be considered before submitting the revision.

1. The authors mentioned that 190/230 patients were sputum smear as well as culture positive. The details of the method used in the drug sensitivity assay need inclusion in the manuscript.

2. The authors are advised to revise the information provided under the heading “MDR-TB Associated Risk factors." There appear some mistakes in the data entry.

3. Line no. 119: Author should mention the method of smear microscopy and reference should be quoted for the guidelines & procedures followed.

4. Line no. 129 Reference is missing.

5. Line no. 134- 135 “Microscopic examinations of culture colonies were done to select AFB positive isolates”. This statement needs to be rephrased for more clarity. How the isolates have been differentiated between MTB & NTMs? As per the WHO guidelines, smear microscopy for TB diagnosis is recommended by Auramine O staining (fluorescent microscopy) however the current study followed ZN method. Hence author should justify the use of ZN method for smear microscopy?

6. According to the WHO guidelines for TB diagnosis, sputum samples are needed to be collected for two consecutive days. In the present study, how the sputum samples have been collected? Whether these samples were from day 1 or day 2 or pooled? The author should clearly mention the collection of sputum samples from the patients enrolled.

7. Line no. 137 “AFB positive isolates” is not clear in this statement whether the direct examination of sputum smear or culture positive isolates confirmed by AFB smear microscopy were taken for molecular typing. The author has not mentioned number of AFB negative isolates throughout the study. The AFB negative isolates have to be investigated using PCR based deletion typing and molecular genetic assay and it must be included in the results. Author may provide supplementary file of data for more clarity.

8. Author should provide schematic diagram of study design for more clarity.

9. The data has not been analyzed by comparing the results with culture (gold standard method). Additionally, the present study has not done the phenotypic drug susceptibility testing which is must to validate the results of molecular drug resistance of isolates.

10. Author needs to reanalyze the data keeping culture as gold standard method.

11. In Table 5, rather than putting column “Types of mutation” author should mention the frequency of individual mutations. The table 3 is described in terms of monoresistance, any resistance and MDR, similarly in table 5 author should mention these terms in the column of “Resistance pattern”. The table 5 needs to be modified in more descriptive and informative way such as by mentioning the change in amino acid from wild type to mutant type with codon position.

12. Line no. 284-285 Author stated “……signaling there is also high and low-level INH resistance in the same isolates” on what basis the author has considered the low and high level of resistance?

13. Line no. 367 the author should correct the statement by replacing “ant-TB drugs” with “anti-TB drugs” and “cells” with “isolates”

14. The authors fail to explain the importance of the present study.

15. The study has very less number of samples to be compared to give any conclusion. The data provided in the manuscript is not efficiently analysed to conclude. Most of the claims of this manuscript are insubstantial and lacks quality for publishing in its present form.

16. Novelty is missing in the study.

17. The authors should revise the discussion part thoroughly in the manuscript which seems very weak to support the study.

18. References are missing at the relevant places.

19. The language of the manuscript need thorough revision for brevity.

6. PLOS authors have the option to publish the peer review history of their article (what does this mean?). If published, this will include your full peer review and any attached files.

Reviewer #1: No

Reviewer #2: Yes: Javaid Ahmad Sheikh

---

## [Author Response · Author response to Decision Letter 0]

23 May 2020

Date 22/05/2020

Response to Reviewers

To: - PLOS ONE Reviewer,

Dear, Sir, Madam

We received the document which contains the reviewer's comments from the PLOS ONE with protocol PONE-D-20-08561 on Thursday, April 16/2020 at 9:27 AM. We would like to thank the reviewers for the attentive and thorough reading of this manuscript and for the thoughtful comments and constructive suggestions, which help to improve the quality of this manuscript. We try to jot down the reviewer's comments and respond to the comments accordingly. 

Comment: The authors mentioned that 190/230 patients were sputum smear as well as culture positive. The details of the method used in the drug sensitivity assay need inclusion in the manuscript. 

Response: Comment accepted and the detailed method used in the drug sensitivity assay is added to the manuscript (pp 8)

Comment: The authors are advised to revise the information provided under the heading “MDR-TB Associated Risk factors." There appear some mistakes in the data entry.

Response: Comment accepted and corrected accordingly (pp 12)

 Comment: Line no. 119: The author should mention the method of smear microscopy and reference should be quoted for the guidelines & procedures followed.

Response: Comment accepted and corrected accordingly (pp 6)

Comments: Line no. 129 Reference is missing.

Response: Comment accepted and corrected accordingly (pp 7)

Comment: Line no. 134- 135 “Microscopic examinations of culture colonies were done to select AFB positive isolates”. This statement needs to be rephrased for more clarity. How the isolates have been differentiated between MTB & NTMs? As per the WHO guidelines, smear microscopy for TB diagnosis is recommended by Auramine O staining (fluorescent microscopy) however the current study followed the ZN method. Hence the author should justify the use of the ZN method for smear microscopy?

Response: Comment accepted and the statement is rephrased to provide more information. The typing was done using the RD9 deletion typing to differentiate MTBC from NTMs. Since the study was conducted in the research center we have used the ZN method because ZN is a gold standard and auramine is for program efficiency (pp 6 - 7).

Comment: According to the WHO guidelines for TB diagnosis, sputum samples are needed to be collected for two consecutive days. In the present study, how the sputum samples have been collected? Whether these samples were from day 1 or day 2 or pooled? The author should clearly mention the collection of sputum samples from the patients enrolled. 

Response: Comment accepted and the procedure is further elaborated to avoid confusion. Accordingly, front-loading is well accepted, WHO has modified case definitions; in any case, the study can have its method and two samples are fine with >97% sensitivity (pp6).

Comment: Line no. 137 “AFB positive isolates” is not clear in this statement whether the direct examination of sputum smear or culture-positive isolates confirmed by AFB smear microscopy were taken for molecular typing. The author has not mentioned the number of AFB negative isolates throughout the study. The AFB negative isolates have to be investigated using PCR based deletion typing and molecular genetic assay and it must be included in the results. The author may provide a supplementary file of data for more clarity.

Response: Comment accepted and the statement has been corrected. In this study, there were no AFB negative isolates because all were smear-positive by entry criterion (pp6 - 7).

Comment: The author should provide a schematic diagram of the study design for more clarity.

Response: Comment accepted and corrected accordingly (Fig 1 supplement file)

Comment: The data has not been analyzed by comparing the results with culture (gold standard method). Additionally, the present study has not done the phenotypic drug susceptibility testing which is a must to validate the results of molecular drug resistance of isolates.

Response: Comment accepted and clearly stated in the limitation part of this study 

Comment: The author needs to reanalyze the data-keeping culture as a gold standard method.

Response: The study is not a methods comparison. In any case, all were culture positive and none were PCR negative.

Comment: In Table 5, rather than putting column “Types of mutation” the author should mention the frequency of individual mutations. Table 3 is described in terms of monoresistance, any resistance, and MDR, similarly, in table 5 author should mention these terms in the column of “Resistance pattern”. Table 5 needs to be modified in a more descriptive and informative way such as by mentioning the change in amino acid from wild type to mutant type with codon position.

Response: Comment accepted and corrected accordingly and the tables are completely modified by new formats Table 2 (pp12) and table 4 (pp 14 )

Comment: Line no. 284-285 The Author stated, “……signaling there is also high and low-level INH resistance in the same isolates” on what basis the author has considered the low and high level of resistance? 

Response: Comment accepted and corrected accordingly (pp14)

Comment: Line no. 367 the author should correct the statement by replacing “ant-TB drugs” with “anti-TB drugs” and “cells” with “isolates”

Response: Comment accepted and corrected accordingly (pp 18)

Comment: The authors fail to explain the importance of the present study.

Response: Comment accepted and corrected accordingly (pp 4 – 5)

Comment: The study has a very less number of samples to be compared to give any conclusion. The data provided in the manuscript is not efficiently analyzed to conclude. Most of the claims of this manuscript are insubstantial and lack quality for publishing in its present form.

Response: Comment accepted and corrected accordingly. The sample size determination for the study primarily focuses on the required number of smear-positive pulmonary TB cases. The study was included all presumptive pulmonary tuberculosis cases during the study period (260 cases) from the selected 20 health facility, these health facilities were purposely selected because more TB patients were seen in these health facilities during the study period.

Comments: Novelty is missing in the study.

Response: The novelty of this study lies in the update it gives of the status in Addis Ababa in 2017 demonstrating that there is no rise in RR or MDR-TB.

Comment: The authors should revise the discussion part thoroughly in the manuscript which seems very weak to support the study.

Response: Comment accepted and corrected accordingly (pp 16 )

Comment: References are missing in the relevant places.

Response: Comment accepted and corrected accordingly 

Comment: The language of the manuscript need a thorough revision for brevity.

Response: Comment accepted and corrected accordingly

---

## [Editor Report · Decision Letter 1]

29 Jun 2020

Molecular detection of multidrug resistance pattern and associated gene mutations in M. tuberculosis isolates from newly diagnosed pulmonary tuberculosis patients in Addis Ababa, Ethiopia.

PONE-D-20-08561R1

Dear Dr. Tilahun,

We’re pleased to inform you that your manuscript has been judged scientifically suitable for publication and will be formally accepted for publication once it meets all outstanding technical requirements.

Kind regards,

Hasnain Seyed Ehtesham

Academic Editor

PLOS ONE

Additional Editor Comments (optional):

I have gone through the revised manuscript and also the Authors response to the comments of the Reviewers. The Authors have added a new Fig as a Supplementary figure and also modified existing Table to address the comments of the Reviewers. The take home message about absence of any increase in incidence of RR and MDR cases in Addis Ababa is significant. In my view, the authors have comprehensively revised the manuscript addressing all the comments of the reviewers. I recommend this manuscript for publication.
---

## [Editor Report · Acceptance letter]

24 Jul 2020

PONE-D-20-08561R1 

Molecular detection of multidrug resistance pattern and associated gene mutations in M. tuberculosis isolates from newly diagnosed pulmonary tuberculosis patients in Addis Ababa, Ethiopia. 

Dear Dr. Tilahun:

I'm pleased to inform you that your manuscript has been deemed suitable for publication in PLOS ONE. Congratulations! Your manuscript is now with our production department. 

Kind regards, 

on behalf of

Prof Hasnain Seyed Ehtesham 

Academic Editor

PLOS ONE